# The Current State of Irrigated Soils in the Central Fergana Desert under the Effect of Anthropogenic Factors

Evgeny Abakumov [1,*], Gulomjon Yuldashev [2], Ulugbek Mirzayev [2], Murodjon Isagaliev [2], Guzalkhon Sotiboldieva [2], Sultonkhuja Makhramhujaev [2], Inomjon Mamajonov [2], Zikrjon Azimov [2], Omonjon Sulaymonov [3], Khasanboy Askarov [3], Barchinoy Umarkulova [3], Abror Rahimov [2] and Timur Nizamutdinov [1]

1   Department of Applied Ecology, Faculty of Biology, Saint Petersburg State University, 16 Line of V.O., 199178 Saint Petersburg, Russia
2   Department of Soil Science, Agrarian Joint Faculty, Fergana State University, Fergana 150100, Uzbekistan
3   Department of Technology of Storage and Initial Processing of Agricultural Products, Faculty of Light Industry and Textiles, Fergana Polytechnic Institute, Fergana 150107, Uzbekistan
*   Correspondence: e_abakumov@mail.ru or e.abakumov@spbu.ru

**Abstract:** This article highlights the role of anthropogenic factors in the modern and stage-by-stage development of soils, using the meadow-marsh soils of Central Fergana as an example. Information on the anthropogenic evolution of desert subtropical soils under long-term irrigation is provided. Data on the component composition of readily soluble salts in soils are discussed. It has been revealed that marsh-meadow soils under the influence of long-term irrigation gradually evolve into the irrigated meadow-saz soils of deserts. It is necessary to organize and conduct monitoring, the results of which could be implemented for the selection of agricultural crops, the development of methods of their sowing and planting, and for development of soil protection methods. In the initial period of using hydromorphic soils for irrigation in desert landscapes, there is a decrease in humus and total nitrogen content. The agrogenic transformation of hydromorphic soils under long-term and intensive use leads to significant changes in a number of soil properties. Each region-specific, soil-climatic condition may have its own pattern of soil areal evolution which is closely linked with the geochemical landscapes and the dynamics of the soil fertility. It is therefore necessary to consider the trends of soil transformation and evolution to improve soil fertility.

**Keywords:** anthropogenic factor; morphology; evolutionary development; Calcisol; gypsum; carbonate; water soluble salts; irrigation transformation

## 1. Introduction

The dominance of humans over the environment, particularly on soil and soil cover, has expanded as consciousness and thinking evolved throughout our development as a species. Anthropogenic activities have a strong influence on the gradual evolutionary development of soils [1]. The south of Uzbekistan is overpopulated and strongly affected by urbanization and agriculture. The Fergana valley belongs to three nations—Uzbekistan, Kyrgyzstan and Tajikistan. The current population of this geographical region is over seven million people. Cities and large villages are located in agricultural landscapes with numerous storage and processing facilities for agricultural products. The agriculture in the region is highly dependent on the redistribution of water from the highlands to the lowlands. Fergana city itself is located at an altitude of 580 m, while one of the oldest cities in the world, Osh, is located at an altitude of 963 m. The Fergana Valley region belongs to the mountain-piedmont areas. Irrigation plays a decisive role here, and water deficits affect the hydrological systems of Central Asia. Intensive irrigation of these lands began in the Soviet period and continues to this day. At the same time, there are a number of problems, including geopolitical ones, associated with the use of irrigation canals and

rivers on the border of the three countries; these problems escalated after the collapse of the Soviet Union.

Under the direct influence of anthropogenic factors, the soils of the Central Fergana desert zone, the intensive development of which began in the 1930s to the 1950s, have undergone significant changes; there were significant transformations in their morphology, as well as their agrochemical parameters and salt composition. In Central Fergana, the first scientific-analytical data on soils with interlayers, which, in the local language were called "Arziks", were presented in the works of Menzi and Klavdienko [2–5]. The so called "Arzyk" soils were studied, as they are the most important component of the soil cover of Fergana valley. After their cultivation and involvement in agricultural practices, a number of problems related to the influence of irrigation on farming and the quality of agricultural products arose. Subsequently, "Arzyk" soils have been categorized into separate groups according to their genesis, and their properties and ways of bio-geochemical rehabilitation have been studied. In studies on the genesis and step-by-step development of the soils formed in the territory in recent years, it was noted that these soils were formed and developed under the influence of specific factors [4,5], among which the main ones are the water regime and chemical composition of the water used for irrigation [6–10].

Few investigations of these soils and their anthropogenic dynamics has been published. Nonetheless, the sandy textures of the soils of the Fergana region and soil hardpan or pedolits have been described, as have the soil anthropogenic salinization and methods for modelling this process [11–19], as well as the problem of groundwater quality and dynamics in the context of salinization [19–21]. In Fergana valley, secondary salinization and intensive of erosion are major environmental problems [22]. Thus, this region faces numerous problems arising from the intensive agricultural use of soil cover and irrigation waters. Salinization has led to considerable deterioration of the physical, chemical, and biological properties of the soils, which have changed from automorphous to hydromorphous in terms of their morphology and geochemistry [23–26].

Bockheim and Hartemink identified changes in soil solutions and their properties depending on evolutionary processes in arid soils [27,28]. Over time, newly irrigated gray-brown soils are transformed into gray-brown-meadow soils in the conditions of Tashsakinskoye plateau [29]. Generalized maps of the USA can be used to analyze the state of irrigated evolving soils [30] or for soil classifications [31], which indicate the soil traits based on the relationship between its properties and the expected responses to anthropogenic influences [32]. Under the influence of anthropogenic factors, the soils of the Djizak steppe may be divided into two zones: the first has no modern salt accumulation, while the second is dominated by active salt accumulation [24].

Thus, despite long-term study of the soils of Central Fergana under conditions of anthropogenic impact, in particular, irrigation, information on the temporal dynamics of soils is insufficient. Therefore, the aim of the present study was to analyze the current and past states of irrigated soils in the region.

The following objectives were set in this research: (1) determination of changes in the morphological structure of meadow-saz soils under the influence of anthropogenic factors; (2) determination of changes in the agrochemical and agrophysical properties of virgin and irrigated soils with arzyk-shokh horizons; (3) determination of the influence of natural and anthropogenic factors, including processes of soil salinization and desalinization; and (4) determination of the genesis and key properties and features of soils with arzyk-shokh horizons in different time periods in Central Fergana, Uzbekistan.

## 2. Materials and Methods

The study site is located in the central part of the Fergana valley (Central Asia, Uzbekistan). The location (40.50023 N, 71.40495 E) is 14 km south-west of the Central Fergana water storage facility (Markaziy farg'ona suv ombaru) (Figure 1). Parent materials are loess type loams with variable carbonate contents. Typical zonal soils are Calcisols (WRB, 2015) or Serozems and Solonchaks (Russian Soil Taxonomy, 2004).

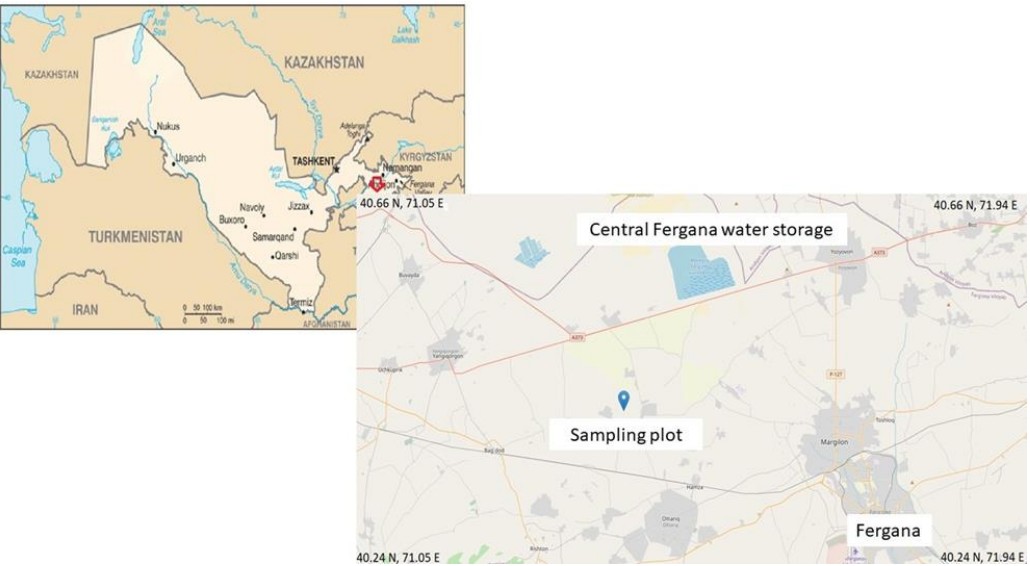

**Figure 1.** Map of soil locations.

These soils are typical for the subtropical ecosystems of the Fergana valley. Soil sections were located on two adjacent arable fields: Sections 1 and 2 were located in non-irrigated fields, while Sections 3 and 4 were located in irrigated fields. The profiles of such soils before irrigation presented by following characteristics (Figure 2): PJ, BCAmc, BCAnc, Cca. Profile 1: PJ (0–37 cm); BCAmc (37–69 cm); BCAnc (69–87 cm); Cca (94–128 cm); Cca (128–154 cm); Cca (155–210 cm); and Profile 2: PJ (0–32 cm); BCAmc (32–45 cm); BCAnc 45–110 cm); Cca (110–142 cm); Cca (142–192 cm); Cca (192–242 cm). The designations are given according to Russian soil taxonomy; thus, PJ is arable (current or former), BCA indicates the secondary accumulation of carbonates with affixes "mc" (diffusive carbonate) and "nc" (nodules of carbonates). If C (parent material) has the designation "ca", it indicates lithogenic, but not pedogenic carbonates. The colors of soil horizons are described as follows: 10 YR 8/4 (PY), 7.5 YR 8/6 (BCA), and 10 YR 8/8 (C). Soils affected by long-term irrigation are presented as follows: PJ, BCA, Cca. Profile 3: PJ (0–30 cm); BCA (30–44 cm); BCA (44–90 cm); Cca (90–136 cm); Cca (136–174 cm); Cca (174–196 cm); and Profile 4: PJ (0–36 cm); BCA (36–70 cm); Cca (70–104 cm); Cca (104–136 cm); Cca (136–171 cm). The only difference in the profile formula is that the secondary accumulation of carbonates does not have diffusive or nodule accumulation thereof, presented here by microparticles in fine earth. This is the result of the saturation of soils by water via irrigation. The color of whole soil became darker due to the peptization of fine particles, i.e., 10 R 2/5 (PY), 7.5 YR 4/5 (BCA) and 5 Y 7/1. Thus, irrigation affects the soil morphology, structure, and color. In general, soil samples were taken from the mountain foothills, i.e., the region with the most productive soil in Uzbekistan. Further north, there are deserts where soils are mostly sandy or strongly enriched with gypsum. There, we sampled four soil sections: two were located on the peripheries of the escape cone irrigated arzyk-gypsum meadow saz soils (soil Sections 1 and 2) and two were located on a plain in the central part of an arable plot, comprising irrigated saline arzyk meadow saz-soils (soil Sections 3 and 4). Soil samples were taken from each soil genetic type. Photos of typical soils before and after irrigation are shown in Figure 2 (Sections 1 and 3).

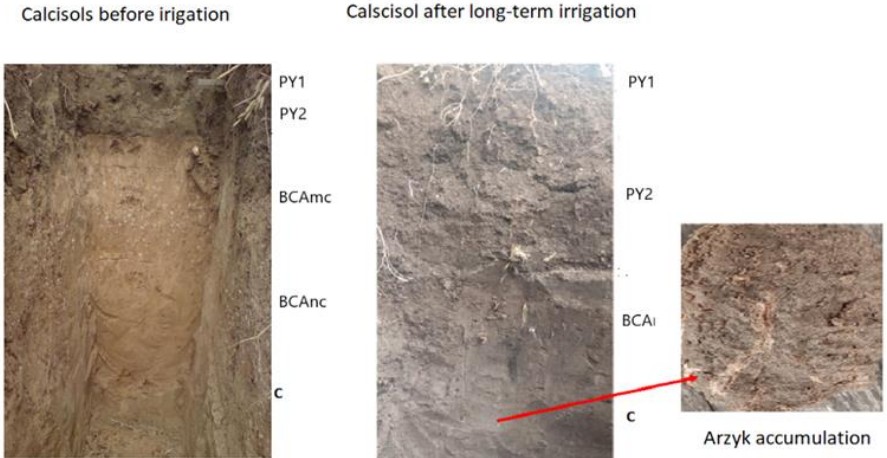

**Figure 2.** The morphology of two typical soil profiles.

Soil samples were air-dried, ground, and passed thorough a 2 mm sieve. Total organic carbon (TOC) was determined by a standard oxidation method [33], and total nitrogen was determined by the standard Kjeldahl Method [34]. To determine the water-soluble salt content, we used the traditional method of analysis of aqueous extract at a ratio of soil:water of 1:5. Dry residue was determined by evaporating 50 mL of the extract and further drying to a constant weight at 105 °C with a gravimetric finish. The calcium and magnesium contents were determined by titration in the presence of Trilon-B (EDTA disodium salt dehydrate). The sodium content was calculated from the difference in the abundances of anions and cations. The chlorine content was determined argentometrically by titration. The potassium and sodium contents were determined by flame-photometry [34]. The particle size distribution of the soils was measured by the sedimentation (gravimetric) method [35]. One of the reasons for using routine methods was that our current data had to be comparable with those in previous studies (in 1975 and 2003). Statistical processing of the data was performed using correlation analysis and calculations of the basic soil characteristics (RMS deviation, coefficient of variation, etc.). Data visualization was performed using GraphPad Prizm 9.0.0 and Golden Software Grapher 13.0.

## 3. Results and Discussions

The soils studied revealed significant changes in morphological organization due to the irrigation effect. The yellowish-pale profiles of Calcisols showed gray streaks and the formation of dense hardpan (arzyk-shokh) in the middle part of the soil profile. Changes in environmental conditions lead to changes in the agro-bio-geochemical properties of soils due to evolutionary processes [36,37] and, therefore, to changes in the genetic horizons of the soil section and its natural structure and properties, leading to an increase in the level of natural fertility. In accordance with this, the properties of soils can be divided into the two following groups: residual properties inherited from the parent rocks—lithorelicts [38] or acquired in the process of soil formation; or properties arising during the development of soils during the stagnation of environmental conditions [1]. The morphological features which are characteristic of the previous stage begin to gradually disappear or transform in the soil profile and new features corresponding to the second stage of soil formation emerge. The grayish color of the soil may therefore be seen as an attribute of soil irrigation process.

Soil-forming factors are unequal in significance but equal in certain respects, i.e., in the degree of their influence. For example, based on the geographical scale of their impact, they may be divided into two groups. The first group includes factors affecting vast areas of land on a broad geographic scale and with different climates, organisms, and the age of the region. The correlation of these factors determines the pattern of the geographical location of significantly different groups of soil types. Thus, geographically, the studied soils were largely subtropical Calcisols which has undergone changes due to intensive irrigation.

Factors for the second group include the parent materials, topography, microclimate, and anthropogenic effect. These factors have different effects on the occurrence of intraspecific soil-forming divisions into smaller taxonomic units within a soil type. These factors manifest in a grayish color and the formation of hardpan material. Ecological factors that have been in equilibrium for centuries in Central Fergana have been disturbed by anthropogenic activities, and the soil cover, which is in a protected state, has undergone extensive changes.

Such changes, especially in Central Fergana, may easily be traced as follows:

(1) Surface planning of rugged terrains in a natural state—agrogenic transformation of relief;
(2) Land irrigation, introduction of irrigation water into agro-ecosystems, which leads to the deposition of limestone-silt sediments and salts;
(3) Changes in the water–salt regime due to the construction of collector–drainage networks;
(4) Complex measures, such as land treatment systems, the replacement of natural vegetation by crops in crop rotation, the implementation of fertilizer systems;
(5) Flush irrigation and other similar hydromeliorative and agromeliorative measures have been implemented;
(6) Secondary salinization that increases arable density;
(7) The initial stage of drainage processes on newly-developed lands is accelerated, etc.

All these procedures essentially effect the soil vertical and special organization in historically irrigated fields in Central Fergana. The irrigated soils of Central Fergana, selected by us to study changes in the genetic layers of the soils, are mainly typical meadow-swamp soils, formed after the 1930s to the 1950s. These sections differ from those of the meadow–swamp soils of the region by the presence of separate specific strata, gypsum, and arzyk-shokh horizons, as well as the appearance, structure, quantity, morphology, and mezomorphology of these formations and their agrochemical properties (Table 1).

**Table 1.** Content of total organic carbon, nitrogen, phosphorus, and potassium in surface and transitional soil horizons (in the lower horizons, the content was below the measurement limits).

| Soil Section | Depth, cm | Horizon | TOC, % | C/N | Bulk Content, % | | |
|---|---|---|---|---|---|---|---|
| | | | | | N | $P_2O_5$ | $K_2O$ |
| | 0–37 | PJ | 0.94 | 9.40 | 0.10 | 0.27 | 1.28 |
| 1 | 37–69 | BCAmc | 0.48 | 9.60 | 0.05 | 0.15 | 1.02 |
| | 69–87 | BCAnc | 0.24 | 12.00 | 0.02 | 0.04 | 0.48 |
| | 0–32 | PJ | 1.37 | 10.53 | 0.13 | 0.29 | 1.20 |
| 2 | 32–45 | BCAmc | 0.62 | 6.89 | 0.09 | 0.19 | 0.96 |
| | 45–110 | BCAnc | 0.13 | 6.50 | 0.02 | 0.09 | 0.51 |
| | 0–30 | PJ | 0.95 | 6.78 | 0.14 | 0.27 | 1.44 |
| 3 | 30–44 | BCA | 0.50 | 5.55 | 0.09 | 0.18 | 0.98 |
| | 44–90 | BCA | 0.19 | 6.33 | 0.03 | 0.10 | 0.63 |
| | 0–36 | PJ | 0.83 | 6.38 | 0.13 | 0.24 | 1.38 |
| 4 | 36–70 | BCA | 0.63 | 7.00 | 0.09 | 0.17 | 1.11 |
| | 70–104 | Cca | 0.24 | 6.00 | 0.04 | 0.14 | 0.57 |

The total carbon content was relatively low compared to our data regarding the Calcisols of Russia or Kazakhstan; the Calcisol of Central Fergana is considered to be the most humus enriched in Uzbekistan. Thus, the soils of the Djizak steppe are very poor in organic carbon. The nitrogen content, i.e., another index of the soil humus state, was very low in terms of gravimetric content, although the C/N ratio was narrow, which is typical for

Serozems (Calcisols) in subtropical climates [39]. There was some biological accumulation of potassium and phosphorous. The structure and morphological indicators were very different from those of other soils (Calcisols without irrigation) in the region. In the soil cover, there was a distinct difference in the genetic horizons and chemical element ratios based on their migration and accumulation. In meadow–swamp and irrigated meadow-saz soils, in the lower part, calcium and magnesium carbonates had accumulated; in the middle, gypsum and carbonates; and in the top, gypsum and water-soluble salts. The accumulation of these compounds comprised tiers, i.e., horizons which were characteristic for them. This arzik-shokh soil cover also had a three-layer or three-tier structure. The topsoil layer consisted of fine-grained soil mass from families of fine-grained and coarse-grained soils, which may contain up to 10% gypsum. At the same time, in the category of surface-sandy soils, the upper part of the section consisted of a gypsum layer containing 20–30% or even more gypsum. The layer also contained water-soluble salts, which tended to vary in concentration from weak to strong.

The middle layer of the section of arzyc soils consisted of two, three, or more shallow layers with hardpan, which may contain 10 to 70% gypsum and 15 to 40% carbonates. The arzyk layers also contained varying amounts of readily soluble salts, some of which were inside the gypsum crystals and hardpan formations. Crushed stone layers were very dense and cemented overlaps of various degrees of strength were often found inside the layer. Their porosity of aeration was very low, and their ability to pass and retain water was poor. The lower part of the section was most often a shokh layer consisting of nodules, i.e., solid or fragmentally cemented and hollow layers. Calcite, dolomite, magnesite, and other carbonate and sulfate minerals accounted for 30–60% of the structural mass of the strata. The virgin Calcisols occurred at the end of the 1970s on the northwestern margins of the cones of the adjacent Isfairam-Shohimardan geomorphological region and in small areas within the irrigated lands of the lake-proluvial plain and in the solonchak complexes [36]. At present, almost all areas characterized by arzyk-shokh soils are included in irrigated agriculture. The eco-ameliorative condition and structure of the soil cover of developed irrigated soils in the studied territory varied within wide limits depending on the duration of the irrigation period, the applied reclamation measures, and the intensity of farming.

Irrigated soils were identified as Solonchaks with hardpan instead of the Calcisols which existed before irrigation. These were preserved until the beginning of the 1980s in the form of spots in small areas within the agro-developed areas [36]. Together with them, significant areas were prepared for development and reclamation or were abandoned due to ineffective development. These soils are characterized by a clayey texture (Figure 2), strong salinity, high carbonate content, and an extremely compacted structure. Owing to the aforementioned influence of anthropogenic factors, the soil profile has undergone complex changes depending on the duration of the development period, as well as on the applied measures 50–70 years after the beginning of the development of arzyk soils and 35–40 years from the time of their primary completion. Active human influence on the natural balance of Central Fergana is largely manifested in intensive capital land planning, construction of sewerage systems and aryks (ditches), as well as collector–drainage and irrigation systems. The main processes manifested in changes in condition related to irrigation and the application of fertilizers. Irrigation played an important role in the activation of soil chemical composition changes, accelerating weathering processes, leaching, and in the redistribution of soil components. All the investigated soils were comparable in terms of texture their class and the details of particle size distribution (Table 2).

**Table 2.** Profile distribution of particle size and physical clay content (<0.01 mm) in soil profiles.

| Section Number | Depth, cm | Fraction Content, % | | | | | | | |
|---|---|---|---|---|---|---|---|---|---|
| | | Horizon | 1.0–0.25 | 0.25–0.10 | 0.10–0.05 | 0.05–0.01 | 0.01–0.005 | 0.005–0.001 | <0.001 | <0.01 |
| 1 | 0–37 | PJ | 1.58 | 12.50 | 19.07 | 38.24 | 15.15 | 7.90 | 5.56 | 28.61 |
| | 37–69 | BCAmc | 6.55 | 17.22 | 14.28 | 16.05 | 13.80 | 23.08 | 9.02 | 45.90 |
| | 69–87 | BCAnc | 11.97 | 23.14 | 10.62 | 21.22 | 11.04 | 16.10 | 5.91 | 33.05 |
| | 94–128 | Cca | 8.03 | 10.22 | 4.14 | 22.35 | 18.42 | 29.65 | 7.19 | 55.26 |
| | 128–154 | Cca | 21.32 | 48.95 | 10.50 | 11.11 | 4.00 | 2.34 | 1.78 | 8.12 |
| | 155–210 | Cca | 3.43 | 15.20 | 18.00 | 36.12 | 8.98 | 9.87 | 8.40 | 27.25 |
| 2 | 0–32 | PJ | 1.39 | 11.97 | 17.29 | 38.17 | 16.60 | 8.78 | 5.80 | 31.18 |
| | 32–45 | BCAmc | 5.10 | 15.80 | 12.90 | 17.91 | 14.54 | 25.38 | 8.37 | 48.29 |
| | 45–110 | BCAnc | 10.85 | 20.32 | 8.87 | 23.22 | 12.60 | 17.02 | 7.12 | 36.74 |
| | 110–142 | Cca | 5.71 | 11.23 | 2.84 | 21.15 | 18.42 | 29.65 | 11.00 | 59.07 |
| | 142–192 | Cca | 23.12 | 45.12 | 8.57 | 10.17 | 4.15 | 2.62 | 1.27 | 8.04 |
| | 192–242 | Cca | 1.33 | 12.29 | 16.23 | 33.07 | 10.75 | 11.87 | 14.40 | 37.02 |
| 3 | 0–30 | PJ | 3.50 | 5.00 | 17.10 | 33.70 | 9.40 | 20.70 | 10.60 | 40.70 |
| | 30–44 | BCA | 5.30 | 5.10 | 26.20 | 32.90 | 13.10 | 18.30 | 9.10 | 40.50 |
| | 44–90 | BCA | 14.90 | 1.50 | 19.10 | 23.40 | 11.50 | 20.30 | 9.30 | 40.10 |
| | 90–136 | Cca | 11.90 | 14.40 | 23.90 | 18.20 | 7.20 | 16.30 | 8.10 | 31.60 |
| | 136–174 | Cca | 8.90 | 9.70 | 16.30 | 24.40 | 8.30 | 13.30 | 19.10 | 40.70 |
| | 174–196 | Cca | 4.10 | 4.60 | 30.10 | 36.10 | 8.10 | 10.20 | 6.80 | 25.10 |
| 4 | 0–36 | PJ | 4.34 | 6.15 | 22.83 | 31.18 | 11.40 | 14.70 | 9.40 | 35.50 |
| | 36–70 | BCA | 3.33 | 5.12 | 22.04 | 26.23 | 14.16 | 20.63 | 8.49 | 43.28 |
| | 70–104 | Cca | 11.27 | 3.90 | 15.04 | 21.30 | 12.08 | 23.30 | 13.11 | 48.49 |
| | 104–136 | Cca | 7.68 | 13.40 | 10.34 | 15.02 | 9.00 | 18.96 | 25.60 | 53.56 |
| | 136–171 | Cca | 6.34 | 7.38 | 14.40 | 20.75 | 11.00 | 18.23 | 21.90 | 51.13 |

At the same time, we observed the accumulation of physical clay in topsoil and in the middle part of the soil profile (Figure 3). This may have been the result of soil infiltration and so-called colmatage, i.e., the accumulation of fine particles in soil porous media during irrigation. This process is quite typical for irrigated soils in all natural zones [40,41]. Colmatage is one of the reasons for the increase in soil density and the formation of hardpan as a specific geochemical barrier. Thus, carbonate and gypsum accumulation occurs not only due to chemical processes, but also due to presence of mechanical geochemical barriers.

Irrigated soils from the very beginning of development were subjected to intensive changes. As a result, under the influence of irrigation loads and cultivation of the surface (30 cm) layers, agro-irrigation arable humus horizons began to form. Today, changes in the structure of the soil cover have largely shifted towards weakening the factors of equilibrium. Further, the rate of soil transformation has changed and now occurs more intensively in them than in mature Calcisols. The scale of these changes will be expressed in, for example, changes in the forms of individual components in the structure of the soil profile within the next 30–40 years. In particular, a high degree of complexity in the thickness of the gypsum layer depth horizon was observed. It was noted that the reason for this is, among others, earth levelling mechanical work. The influence of irrigation on gypsum forms is clearly expressed in soil sections. The gypsum layer, consisting of thin and small crystals of gypsum and its derivatives, is usually located close to the ground surface. The size of the crystals increased with an increase in the depth of the soil section, which started to take the form of a rhombohedrum as the depth increased. In gypsum layers, the phenomenon of suffusion was clearly expressed, indicating gypsum leaching. This phenomenon may intensify over time.

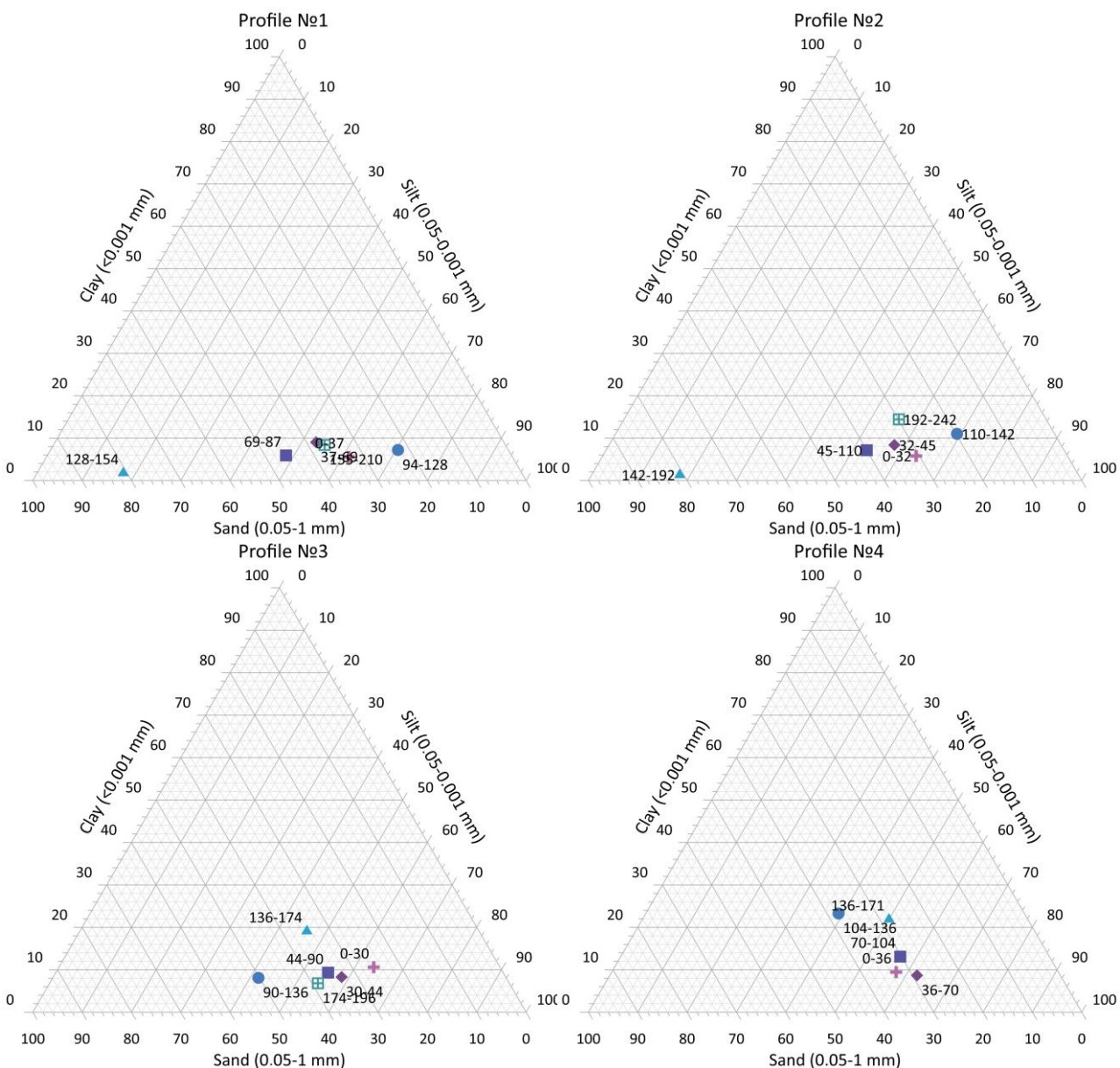

**Figure 3.** Particle size distribution diagrams.

In the fine-grained gypsum layer, the amount of gypsum in the natural state was much higher and the color was white. Irrigation water, affecting in the process of irrigation from top to bottom, unloads fine-grained soil mass from the arable layer to the gypsum layer, depositing the former in the spaces between gypsum crystals. From the gypsum layer, it partially dissolves gypsum crystals, causing them to sink to the bottom and washing out the small crystals [42–44]; the longer the duration of watering, the greater the extent of this process. As a result, the snow-white gypsum layer turns into a fine-grained gypsum-clay mixed layer with a cloudy, whitish-gray tint. Under the influence of irrigation, tillage, and other measures, the arable soil layer became homogeneous in structure. Gypsum formations partially dissolved and the size of the remaining crystals became even smaller. These crystals were then mixed with the soil and difficult to discern. At the same time, the content of organic matter gradually increased. For this reason, the roots of plants penetrated into the arable, subsoil horizons, which are significantly more abundant in the upper part of the layer, sharply decreasing toward the horizons containing gypsum. There were small roots in the gaps between the structural fragments. In this layer, salt leaching phenomena were found to be more pronounced. In the lower strata of the sections,

the effects of suffusion processes could also be observed, although these were weak and not found everywhere. The Syr-Darya river and its tributaries, Naryn and Kara-Darya, as well as the mountain rivers flowing from the mountain ridges surrounding the valley, are the source of irrigation water for the surrounding cultivated lands. The flow of these waterways results in erosion and colmatation (infiltration of soil porous layer with fine particles transported by water).

The average annual water flow of the Naryn river (a tributary of the Syr-Darya River) near Uchkurgan city is 405 m$^3$/s of sediment, taking into account the waters of the Big Fergana Chamel (BFC). The average flow rate in the river water is 408 kg/s. The average annual volume of river sediment reaches 12,854 thousand tons. The average turbidity of the river water is 1.09 kg/m$^3$, and the volume of wash mass per 1 km$^2$ of the catchment area is 241 t/year. The average annual flow of the Kara-Darya River (a tributary of the Syr-Darya) varies within the range of 69.6–207 m$^3$/s. Its average annual flow rate at the Kampirovat site is 118 m$^3$/s. The quantity of solid particles averages 200 kg/s, amounting to 6320 tons per year. The average turbidity of the river water is 1.64 kg/m$^3$ and the volume of solid runoff per 1 km$^2$ of the catchment area is 511 t/year. The washing activity of mountain rivers flowing from the Turkestan and Alay Ridges is also significant. The amount of solid runoff for each km$^2$ of the catchment area of the Sokh river is about 500 t/year; that of the Isfara river is about 200 t, that of the Shakhimardan about 100 t, that of the Akbura about 46 t, and that of the Isfahan about 39 t [12]. The thickness of the agro-irrigation layer in the soils of Fergana valley is, on average 60–80 cm. In addition to a history of many thousands of years of valley irrigation, this figure also illustrates the erosion and slope of the land surface. According to [4,5], the thickness of the agro-irrigation layer reaches 1–1.5 m in many parts of the valley. The formation of the agro-irrigation layer also depends on the location of the irrigation system, irrigation intensity, as well as the quantities of various substances (old wall residues, silt, organic waste, etc.) introduced into the soil. On the irrigated lands under the management of the Big Fergana Chanel (BFC), the arable layer in Arzic soils, formed 40–60 years ago, is clearly distinguishable by the changes that have occurred due to irrigation. In the arzic soils irrigated by the Big Andijan Chanel (BAC), the arable layer has changed little over the years. The water there has almost no sediment load and is relatively transparent. The BFC water is turbid; its volume in early spring is 0.590 kg/m$^3$, and the amount of solid sediments during the growing season exceeds 1–1.5 kg/m$^3$. The soils described above are characterized at that time by high contents of readily soluble salts. Therefore, in 1903–1904, the total content of readily soluble salts in these soils was 1.5% in the top horizons, 9% in the median (on depth up to 70 cm) layer, and 30% at a depth 100–110 cm, which is certainly not typical for all soils in the Fergana valley. Of course, this phenomenon occurs only in accumulative irrigated landscapes.

Nowadays, in the slightly saline soils of the studied massif, the dry residue content averages 0.5–0.7%, and the maximum values of this indicator are in the arable layer in the autumn. In the lower boundary parts of the cone, mainly strongly saline and saline soils with high contents of water-soluble salts in the upper layers prevail, while the dry residue content in some sections reaches 60%. The total amount of salts showed a decreasing trend with depth; these compounds clearly accumulated in the gypsum and arsic horizons. At the same time, the salt content decreased in the lower shochemical part of the section. In our study, the results of an aqueous extract analysis (Table 3) showed that 25 years of irrigation on the distribution of salts in the soil had little effect, but, at the same time, that salinity persists throughout the soil profile.

In some layers, the abundances of both total salts and phytotoxic salts were found to have increased; in other words, redistribution had taken place. Additionally, the salinity type became stable, i.e., sulfate–calcium. The pH of the soil fine earth was close to neutral due to the presence of normal carbonates [39]. The pH of soils under sodium salinization should be more than 8.5. As such, the soils of Central Fergana are characterized by normal carbonate salinization. There were no essential changes in the gypsum state of the soil (Table 4, Figure 4), and thus, there has been a stable development of the soil from 1975 until the present.

**Table 3.** The composition of water-soluble salts in soil profiles.

| Soil Section | Depth, cm | Horizon | Dry Residue,% | pH | CaCO$_3$ | HCO$_3^-$ | Cl$^-$ | SO$_4^{2-}$ | Ca$^{2+}$ | Mg$^{2+}$ | Na$^+$ |
|---|---|---|---|---|---|---|---|---|---|---|---|
| | | | | | % | | | | | | |
| 1 | 0–37 | PJ | 1.24 | 7.10 | 4.46 | 0.018 | 0.016 | 0.840 | 0.240 | 0.025 | 0.055 |
| | 37–69 | BCAmc | 1.28 | 7.12 | 2.65 | 0.021 | 0.016 | 0.888 | 0.230 | 0.025 | 0.132 |
| | 69–87 | BCAnc | 1.19 | 7.15 | 2.52 | 0.015 | 0.023 | 0.792 | 0.234 | 0.015 | 0.103 |
| | 94–128 | Cca | 1.24 | 7.20 | 4.35 | 0.012 | 0.016 | 0.768 | 0.206 | 0.025 | 0.099 |
| | 128–154 | Cca | 1.02 | 7.20 | 12.95 | 0.009 | 0.008 | 0.696 | 0.206 | 0.002 | 0.101 |
| | 155–210 | Cca | 0.30 | 7.25 | 18.29 | 0.018 | 0.017 | 0.144 | 0.029 | 0.002 | 0.043 |
| 2 | 0–32 | PJ | 1.44 | 7.15 | 4.10 | 0.029 | 0.040 | 0.926 | 0.288 | 0.050 | 0.056 |
| | 32–45 | BCAmc | 1.22 | 7.15 | 3.20 | 0.025 | 0.015 | 0.835 | 0.260 | 0.029 | 0.063 |
| | 45–110 | BCAnc | 1.19 | 7.15 | 3.40 | 0.022 | 0.009 | 0.800 | 0.260 | 0.023 | 0.054 |
| | 110–142 | Cca | 1.08 | 7.25 | 5.90 | 0.021 | 0.009 | 0.731 | 0.210 | 0.016 | 0.093 |
| | 142–192 | Cca | 1.16 | 7.20 | 11.80 | 0.018 | 0.012 | 0.794 | 0.254 | 0.018 | 0.069 |
| | 192–242 | Cca | 0.30 | 7.22 | 16.40 | 0.012 | 0.015 | 0.183 | 0.056 | 0.009 | 0.024 |
| | 242–270 | Cca | 0.71 | 7.50 | 16.95 | 0.013 | 0.009 | 0.481 | 0.138 | 0.029 | 0.022 |
| 3 | 0–30 | PJ | 1.35 | 7.20 | 1.25 | 0.025 | 0.026 | 0.904 | 0.295 | 0.047 | 0.030 |
| | 30–44 | BCA | 1.24 | 7.30 | 2.30 | 0.015 | 0.018 | 0.848 | 0.272 | 0.042 | 0.030 |
| | 44–90 | BCA | 1.20 | 7.30 | 2.45 | 0.015 | 0.018 | 0.804 | 0.264 | 0.039 | 0.025 |
| | 90–136 | Cca | 1.10 | 7.35 | 5.36 | 0.017 | 0.016 | 0.748 | 0.252 | 0.027 | 0.035 |
| | 136–174 | Cca | 1.12 | 7.50 | 8.80 | 0.018 | 0.013 | 0.752 | 0.248 | 0.023 | 0.046 |
| | 174–196 | Cca | 0.32 | 7.80 | 8.82 | 0.012 | 0.012 | 0.201 | 0.064 | 0.006 | 0.023 |
| 4 | 0–36 | PJ | 0.59 | 7.20 | 2.15 | 0.010 | 0.010 | 0.386 | 0.134 | 0.011 | 0.021 |
| | 36–70 | BCA | 0.62 | 7.25 | 2.56 | 0.012 | 0.013 | 0.408 | 0.142 | 0.013 | 0.022 |
| | 70–104 | Cca | 0.82 | 7.30 | 2.85 | 0.014 | 0.016 | 0.539 | 0.180 | 0.021 | 0.028 |
| | 104–136 | Cca | 1.24 | 7.50 | 3.40 | 0.015 | 0.016 | 0.842 | 0.256 | 0.050 | 0.031 |
| | 136–171 | Cca | 1.07 | 7.8 | 5.60 | 0.015 | 0.014 | 0.736 | 0.216 | 0.047 | 0.030 |

**Table 4.** Gypsum content in soils: temporal dynamics.

| Soil Section | Depth | CaSO$_4$, % | | |
|---|---|---|---|---|
| | | 1975 | 2003 | 2015 |
| 1 | 0–37 | 21 | 22 | 20 |
| | 37–69 | 45 | 46 | 45 |
| | 69–87 | 70 | 69 | 67 |
| | 94–128 | 58 | 57 | 55 |
| | 128–154 | 33 | 33 | 29 |
| | 155–210 | 2 | 2 | 1 |
| 2 | 0–32 | 9 | 10 | 11 |
| | 32–45 | 29 | 26 | 27 |
| | 45–110 | 54 | 49 | 49 |
| | 110–142 | 53 | 51 | 50 |
| | 142–192 | 65 | 63 | 56 |
| | 192–242 | 44 | 42 | 34 |
| 3 | 0–30 | 9 | 8 | 10 |
| | 30–44 | 26 | 25 | 24 |
| | 44–90 | 47 | 45 | 45 |
| | 90–136 | 54 | 52 | 53 |
| | 136–174 | 61 | 61 | 59 |
| | 174–196 | 41 | 41 | 39 |
| 4 | 0–36 | 7 | 7 | 6 |
| | 36–70 | 13 | 12 | 12 |
| | 70–104 | 16 | 16 | 15 |
| | 104–136 | 16 | 16 | 17 |
| | 136–171 | 19 | 19 | 18 |

Correlation analyses (Figure 5) showed that sulfate anions and calcium cations had maximal correlations with dry residue. In addition, there was strong correlation between sulfate and calcium. The lower and negative correlation is characteristic for magnesium and sodium ions, as these cations are parts of opposite trends of salinization.

Mathematical treatment showed that the arithmetic mean dry salt content in all horizons of all sections was 1.002% RMS deviation $\pm$ 0.34, with a coefficient of variation $\pm$ 33.59. For Sections 1, 3, 4 the arithmetic mean content of dry salt residue averaged 0.99% RMS deviation $\pm$ 0.33 with a coefficient of variation $\pm$ 33.71. As indicated above, under the influence of washing and irrigation waters, and in connection with the transition of the water regime away from a stable-marsh regime, the abundances of soil salts sharply decreased, although high enough degrees of salinization remained. As a result, by the 1970s, high salinity levels were observed in the gypsum-arsyk soils, as well as in all soils of the region. However, irrigated gypsum-arsyk meadow-saline loamy soils remained saline according to 1975–1978 data. The arithmetic mean gypsum contents in Transects 1, 3, 4 were 39.04% mean square deviation is $\pm$18.9, with a coefficient of variation $\pm$ 48.42.

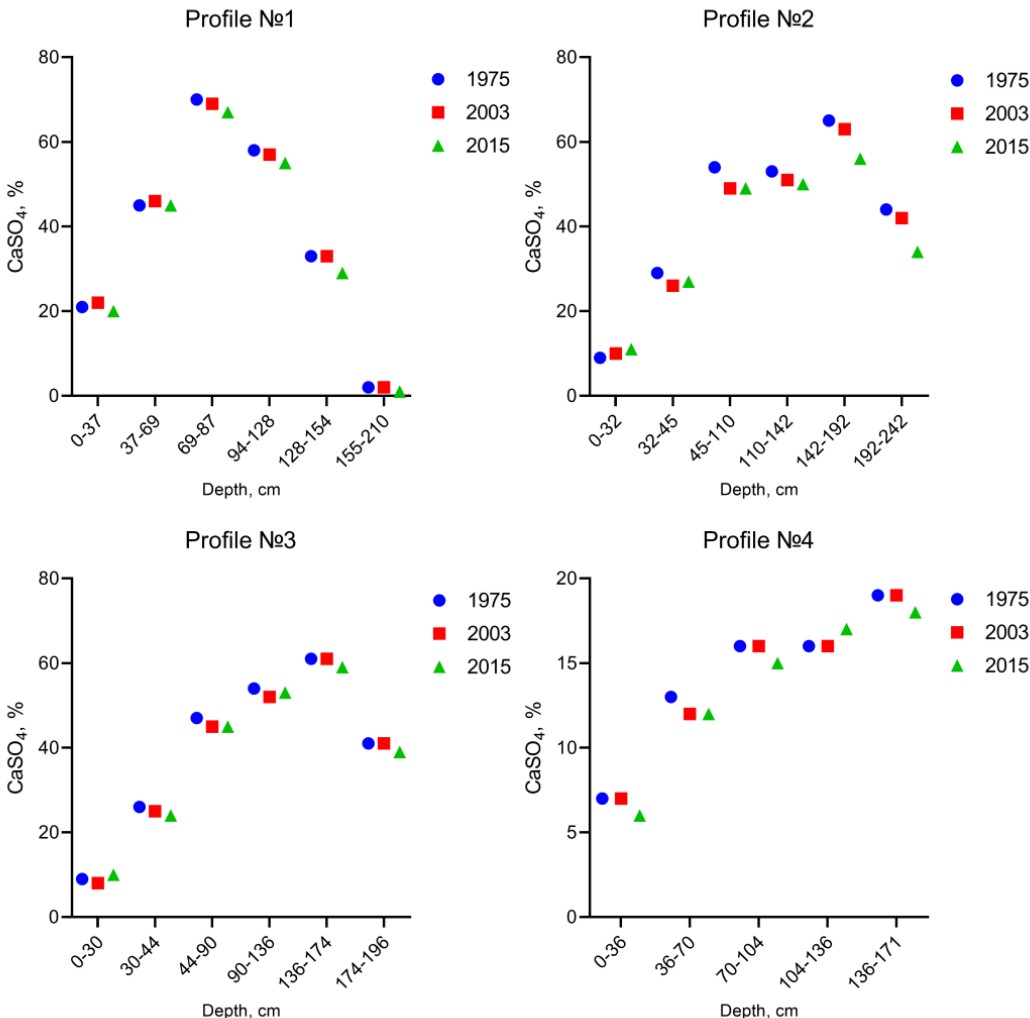

**Figure 4.** Distribution of gypsum content in soil profiles from 1975 to 2015.

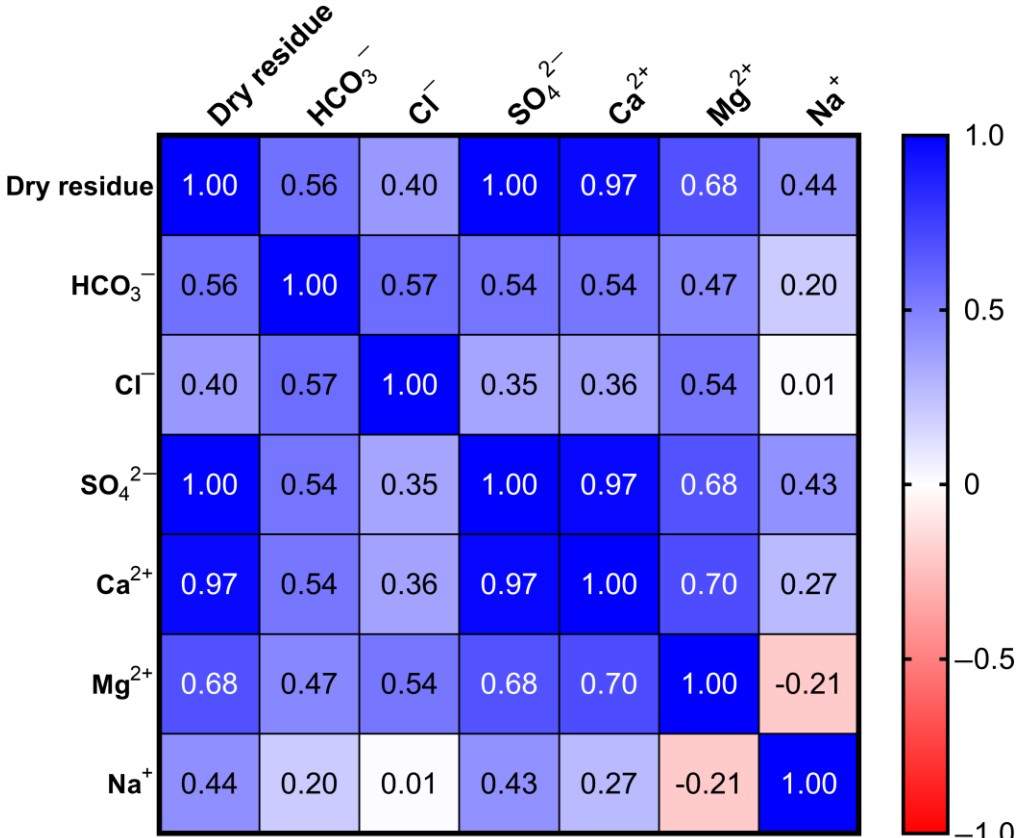

**Figure 5.** Correlation matrix (Pearson r) of the dynamics of water-soluble salt contents.

### 4. Conclusions

Anthropogenic development in the Central Fergana valley has led to radical transformation of the soil water regime and salt composition. When applying irrigation, it is important to take into consideration the relief of the area and the specificities of the flora. In the ontogenetic evolution of soils with arzyk-shokh horizons, especially when irrigation is applied, anthropogenic factor are known to regulate key evolutionary processes. In the initial period of using hydromorphic soils for irrigation in desert landscapes, there is a decrease in humus and total nitrogen content. According to the results of the present study, we can say that the soil profiles of "Arzyk" have been clarified and previously accumulated carbonates and gypsum have disappeared as a result of hydromorphism. The soil chemical properties have also changed; processes of dehumification and reduction of C/N ratio (from 9.4 to 6.4 under prolonged irrigation) were noticed. The contents of bulk phosphorus and potassium are changing insignificantly. The process of the removal of clay particles deep in the profile was also observed. Finally, an increase in the physical clay (<0.01 mm) content by 9–10% was observed. A significant relationship (r > 0.9) in the dynamics of the concentrations of water-soluble calcium salts was detected, which also confirmed our hypothesis about salt removal as a result of hydromorphism. The agrogenic transformation of hydromorphic soils under long-term and intensive use leads to significant changes in a number of soil properties. In general, we can conclude that each region-specific, soil-climatic condition may have its own pattern of soil spatial evolution which is closely linked with the geochemical landscape and the dynamics of the soil fertility. Consideration of the trends of soil transformation and evolution are necessary for the development of methodologies to improve soil fertility.

**Author Contributions:** E.A. conceptualization and methodology; G.Y. methodology; U.M. writing; M.I. visualization; G.S. data curations; S.M. data curation; I.M. original draft preparation; Z.A. filed survey; O.S. laboratory research; K.A. writing; B.U. filed survey; A.R. writing; T.N. software, statistics. All authors have read and agreed to the published version of the manuscript.

**Funding:** This work was supported by international project "Development of theoretical and practical basics of soil and plant geochemistry in Fergana Valley" in 2013–2018. This work was also supported by the Ministry of Science and Higher Education of the Russian Federation in accordance with agreement No. 075-15-2022-322 date 22 April 2022 on providing a grant in the form of subsidies from the Federal budget of Russian Federation. The grant was provided for state support for the creation and development of a World-class Scientific Center "Agrotechnologies for the Future".

**Institutional Review Board Statement:** Not applicable.

**Informed Consent Statement:** Not applicable.

**Data Availability Statement:** Data are provided by the authors on request.

**Acknowledgments:** This research was partially supported by scientific equipment of Scientific park of Saint-Petersburg State University, Chemical Analysis and Materials Research Centre and Environmental Safety Observatory.

**Conflicts of Interest:** The authors declare no conflict of interest.

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
