# Peer review of "The Current State of Irrigated Soils in the Central Fergana Desert under the Effect of Anthropogenic Factors"

_geosciences, doi:10.3390/geosciences13030090_

Round 1

Reviewer 1 Report

This is an interesting article that examines the effects of irrigation and anthropogenic factors in the soil degradation of the agriculturally important Fergana Valley.

I found the article interesting and I believe the community would find value in this work. I would recommend moderate-to-minor edits. My main issue is that the conclusions and discussion need some more explanation. How do we reach the conclusions the authors made from the results? This should be better explained. 

- I found many minor typos (i.e. "Fergana valley belong" instead of belongs in line 36) and expression mistakes.
As non-exaustive examples, line 86 "this region faced to numerous problems", line 147 "This is resulted from", L168 "our current data must be comparable and comparable", L172 "iunder" etc

The expression mistakes make parts of Section 3 incomprehensible as there are sentences that end abruptly (L268-269 and other cases). 

- I believe the introduction would benefit if the historical context and information was somewhat reduced to more relevant parts.

- In the materials it should be noted when the samples were gathered (year and season). 

- I don't understand what you are trying to say in L256 with "Sandy Massives"

- The conclusions should be expanded. There are conclusions stated that are not well-related to the study. Please expand in the discussion how the study findings could help with the selection of crops before mentioning it in the conclusions.
The novelty of the study should be further highlighted in the conclusions, not just in the introduction.
There should be more direct reference to the four goals set in the introduction and whether they were achieved satisfactory.  

Author Response

Dear anonymous reviewers!

Thank you for your attention to our article and your valuable comments! We have taken them all into consideration and tried to improve the article.

Below is a list of responses to your comments

Review 1

- I found many minor typos (i.e. "Fergana valley belong" instead of belongs in line 36) and expression mistakes.
As non-exaustive examples, line 86 "this region faced to numerous problems", line 147 "This is resulted from", L168 "our current data must be comparable and comparable", L172 "iunder" etc

Corrected

The expression mistakes make parts of Section 3 incomprehensible as there are sentences that end abruptly (L268-269 and other cases). 

Corrected

- I believe the introduction would benefit if the historical context and information was somewhat reduced to more relevant parts.

The introduction has been shortened and structured

- In the materials it should be noted when the samples were gathered (year and season). 

Information has been added.

- I don't understand what you are trying to say in L256 with "Sandy Massives"

Corrected

- The conclusions should be expanded. There are conclusions stated that are not well-related to the study. Please expand in the discussion how the study findings could help with the selection of crops before mentioning it in the conclusions. The novelty of the study should be further highlighted in the conclusions, not just in the introduction. There should be more direct reference to the four goals set in the introduction and whether they were achieved satisfactory.  

The conclusion section has been supplemented in accordance with the objectives of the study

Reviewer 2 Report

The introduction is very long and provides too much information about soil characteristics and processes which are not well organized. It can be more straightforward towards the research questions you aim to answer. Please consider including the following paragraphs only: general geographic background of the study area, a summary of previous studies on soils in this area, the knowledge gap, and the aims and hypotheses of this study. Additionally, the introduction uses lots of long sentences which are difficult to read (e.g., lines 70-75). Some sentences are not complete or have grammatical issues (e.g., lines 64-65, 83-84) which should be solved.

In materials and methods, two typical profiles with and without irrigation were described in detail (lines137-152, Fig. 2). Are these from your sampled four profiles or somewhere else? More information should be provided about this.

For the four profiles you sampled, you labeled them as soil sections 82, 8, 1, and 2. These seem like random numbers to the audience. Why not simply label them as “1, 2, 3, 4”, or “Non-irrigated 1 and 2” and “Irrigated 1 and 2”? Please also provide a map of the detailed profile locations and profiles pictures for these four profiles. How did you select the locations of the four profiles? What is classification of the four soil profiles?

Typo: line 152, “petrogypcis deserts”

Line 160, total carbon or organic carbon?

Line 161, citation for Kjeldahl Method.

What method did you use for measuring particle size fraction?

What data analysis methods or statistical analysis did you use to interpret the results?

Typo: line 173, “iunder”

Table 1 shows fewer horizons than Tables 2, 3, 4. Is that because the TOC, C/N are not detected in deeper soils?

What are the horizon designations of your sampled soils? Please add this in the materials and methods or in the Table.

The captions of the tables are too short. They should be sentences instead of a few words.

The authors may consider plotting some depth functions to visualize the vertical distribution of different properties.

Author Response

Dear anonymous reviewer!

Thank you for your attention to our article and your valuable comments! We have taken them all into consideration and tried to improve the article.

Below is a list of responses to your comments

Review 2

The introduction is very long and provides too much information about soil characteristics and processes which are not well organized. It can be more straightforward towards the research questions you aim to answer. Please consider including the following paragraphs only: general geographic background of the study area, a summary of previous studies on soils in this area, the knowledge gap, and the aims and hypotheses of this study. Additionally, the introduction uses lots of long sentences which are difficult to read (e.g., lines 70-75). Some sentences are not complete or have grammatical issues (e.g., lines 64-65, 83-84) which should be solved.

The introduction has been shortened and structured

In materials and methods, two typical profiles with and without irrigation were described in detail (lines137-152, Fig. 2). Are these from your sampled four profiles or somewhere else? More information should be provided about this.

For the four profiles you sampled, you labeled them as soil sections 82, 8, 1, and 2. These seem like random numbers to the audience. Why not simply label them as “1, 2, 3, 4”, or “Non-irrigated 1 and 2” and “Irrigated 1 and 2”?

The numeration of the profiles has been unified throughout the text of the article.

Please also provide a map of the detailed profile locations and profiles pictures for these four profiles. How did you select the locations of the four profiles? What is classification of the four soil profiles?

Classification has been added. Unfortunately, we do not have photo of all profiles.

Typo: line 152, “petrogypcis deserts”

Corrected

Line 160, total carbon or organic carbon?

Total organic carbon - corrected

Line 161, citation for Kjeldahl Method.

Corrected

What method did you use for measuring particle size fraction?

Method was added and described.

What data analysis methods or statistical analysis did you use to interpret the results?

Paragraph about statistics has been added.

Typo: line 173, “iunder”

Corrected

Table 1 shows fewer horizons than Tables 2, 3, 4. Is that because the TOC, C/N are not detected in deeper soils?

Title corrected. Measurements were taken only in the topsoil and transitional horizons, because C and N content in the parent material is extremely low

What are the horizon designations of your sampled soils? Please add this in the materials and methods or in the Table.

Data has been added

The captions of the tables are too short. They should be sentences instead of a few words.

Corrected

The authors may consider plotting some depth functions to visualize the vertical distribution of different properties.

Plots have been added (Figure 4)

Round 2

Reviewer 2 Report

The manuscript has been improved after the revision.